# Sequence-based prediction of the intrinsic solubility of peptides containing non-natural amino acids

Marc Oeller [1,6], Ryan J. D. Kang [1], Hannah L. Bolt [2], Ana L. Gomes dos Santos [3], Annika Langborg Weinmann [4], Antonios Nikitidis[5], Pavol Zlatoidsky[5], Wu Su [5], Werngard Czechtizky[5], Leonardo De Maria [5], Pietro Sormanni [1] ✉ & Michele Vendruscolo [1] ✉

Non-natural amino acids are increasingly used as building blocks in the development of peptide-based drugs as they expand the available chemical space to tailor function, half-life and other key properties. However, while the chemical space of modified amino acids (mAAs) such as residues containing post-translational modifications (PTMs) is potentially vast, experimental methods for measuring the developability properties of mAA-containing peptides are expensive and time consuming. To facilitate developability programs through computational methods, we present CamSol-PTM, a method that enables the fast and reliable sequence-based prediction of the intrinsic solubility of mAA-containing peptides in aqueous solution at room temperature. From a computational screening of 50,000 mAA-containing variants of three peptides, we selected five different small-size mAAs for a total number of 37 peptide variants for experimental validation. We demonstrate the accuracy of the predictions by comparing the calculated and experimental solubility values. Our results indicate that the computational screening of mAA-containing peptides can extend by over four orders of magnitude the ability to explore the solubility chemical space of peptides and confirm that our method can accurately assess the solubility of peptides containing mAAs. This method is available as a web server at https://www-cohsoftware.ch.cam.ac.uk/index.php/camsolptm.

Peptides are a growing drug market with over 100 approved drugs, with insulin being the most prominent one[1–3]. Peptide drugs exhibit several advantages over small molecules[2]. Since they often exhibit low toxicity and may not accumulate in tissue, they can be safe while having high efficacy[2]. They are also diverse, potent, easy to synthesise[2] and have higher specificity, due to their larger size compared to small molecules[4]. However, peptide drug candidates can suffer from several problems. They tend to have low oral bioavailability and short half-lives[1,2,5] caused by high clearance rates and low metabolic stability due to the presence of peptidases[1,2,5]. Moreover, peptides can have poor

[1]Centre for Misfolding Diseases, Yusuf Hamied Department of Chemistry, University of Cambridge, Cambridge, UK. [2]Hit Discovery, Discovery Sciences, BioPharmaceuticals R&D, AstraZeneca, Cambridge, UK. [3]Advanced Drug Delivery, Pharmaceutical Sciences, BioPharmaceuticals R&D, AstraZeneca, Cambridge, United Kingdom. [4]Early Chemical Development, Pharmaceutical Sciences, BioPharmaceuticals R&D, AstraZeneca, Gothenburg, Sweden. [5]Medicinal Chemistry, Research and Early Development, Respiratory and Immunology, BioPharmaceuticals R&D, AstraZeneca, Gothenburg, Sweden. [6]Present address: Proteomics and Signal Transduction, Max Planck Institute of Biochemistry, Martinsried, Germany. ✉e-mail: ps589@cam.ac.uk; mv245@cam.ac.uk

membrane permeability, tend to aggregate, can contain immunogenic sequences[2,6], and their conformational flexibility may generate problems during drug development as they can adopt more than one structure[5].

Taking example from nature, the properties of endogenous peptides and proteins can be modified through post-translational modifications (PTM)[7]. Typical PTMs include phosphorylation for signal transduction and energy metabolism[8,9], and acetylation and glycosylation for regulation[10]. Other common modifications are amidation, carboxylation, hydroxylation, disulfide bond formation, sulfation and proteolytic cleavage[11,12]. PTM dysregulation is often associated with disease, including sleeping sickness[13], amyloid-associated diseases[14] and HIV[15]. A particular focus in recent years has been put on the impact of PTMs on protein aggregation, and on associated neurodegenerative diseases[6,16]. Different PTMs have been shown to have varying effects on the aggregation propensity of peptides and proteins[6]. N-terminal truncation, incorporation of pyroglutamate, phosphorylation and nitration increases oligomerisation of the amyloid-β peptide, while citrullination and backbone modifications also increase oligomerisation but simultaneously decrease aggregation[6]. In therapeutic applications, examples include the increase in biological activity and improvement of metabolic stability by N-methylation[17,18], increasing binding affinity[4,19], half-life increase and improvement of tissue penetrating abilities by lipidation and acylation[6]. Methylation can also increase binding selectivity[19].

By adopting strategies that extend the scope of PTMs, the use of modified amino acids (mAAs) has become prominent in biotechnology and drug development[3], through a variety of methods to engineer mAAs into proteins[20–29]. A selection of the most common mAAs is shown in Table 1, with those used in this work being highlighted in bold. General approaches to improve peptide-based drugs often start with alanine or glutamic acid scanning to identify interaction and cleavage sites[5], and continue with the replacement of natural amino acids with modified amino acids (mAAs) to tailor a variety of other properties[1,5]. These mAAs can contain new functional groups, and alter the backbone or the terminal structure of a peptide[5,30]. The effects of mAAs are diverse and can counter specific problems inherent in biologics, including by altering immunogenicity[31]. One of the major issues in peptide drug development is the recognition by proteases and peptidases, which can be attenuated by changing the backbone through incorporation of amide bond mimics, D-isomers, β-amino acids, alteration of the termini or tetra-substituted amino acids[1,4,17,19,31–36]. These mimics also tend to increase bioavailability, another issue which often plagues peptide drugs[17] as well as restrict conformation and therefore reduce flexibility[1,37,38]. Similar effects can also be caused by N-alkylations[1,17], incorporation of aminoisobutyric acid[39], other constraining amino acids[31,40,41] or by cyclisation[1,19,36,38]. The latter and addition of sterically bulky groups can also reduce T-cell recognition[4,19]. Bioavailability and stability can also be improved by glycosylation, which enhances protein-protein interactions and makes use of glucose transporters on the cell surface which improves cell permeability[31]. Permeability can also be improved by increasing hydrophobicity, which can be achieved by methylation, lipidation[31], and by adding fluorinated residues[19] or modifications to terminal residues[42].

Many applications based on mAAs have been made in materials science, especially with nanotubes and nanofibres[43–46]. mAAs can be also used for photoactive, photo- or fluorescent-caged and photo-crosslinking modifications[47–56], fluorescent probes[47,48,57–60], spectroscopic probes[47,48,61] and as metal ion chelators[47,48]. Moreover, they can be used to create redox-active enzymes[62], reduce the complexity of NMR spectra[63] and can have antimicrobial activity[64].

Commercial vendors currently offer hundreds of synthesis-ready mAAs that can be synthesised into peptides and it has been shown recently that this chemical space can be greatly expanded[65]. At the same time, experimental methods to characterise peptides are

**Table 1 | Selection of the most common modified amino acids (mAAs)**

| Amino Acid | Modification |
| --- | --- |
| Ala | N-acetylation (N-terminus) |
| Ala | **Aminoisobutyric acid** |
| Ala | **Cyclohexylalanine** |
| Ala | **Addition of a primary amine** |
| Arg | **Deimination to citrulline** |
| Arg | Dimethylation (N, N-Met) |
| Arg | Methylation (O-Met) |
| Arg | Methylation (N-Met) |
| Asn | Deamidation to Asp or iso-Asp |
| Asn | N-linked glycosylation |
| Asp | Isomerization to isoaspartic acid |
| Asp | **N-acetylation (N-terminus)** |
| Cys | Disulfide-bond formation |
| Cys | N-acetylation (N-terminus) |
| Cys | Oxidation to sulfonic acid |
| Cys | S-nitrosylation |
| Gln | Cyclization to pyroglutamic acid (N-terminus) |
| Gly | N-acetylation (N-terminus) |
| His | Phosphorylation |
| Leu | **Norleucine** |
| Leu | **Methylation (tert-Butyl-Alanine)** |
| Lys | Hydroxylation |
| Lys | **Acetylation** |
| Lys | Methylation |
| Lys | Ubiquitination |
| Lys | SUMOylation |
| Met | N-acetylation (N-terminus) |
| Met | Oxidation to sulfoxide |
| Met | Oxidation to sulfone |
| Phe | **C-amidation (C-terminus)** |
| Pro | Hydroxylation |
| Ser | N-acetylation (N-terminus) |
| Ser | O-linked glycosylation |
| Ser | Phosphorylation |
| Thr | N-acetylation (N-terminus) |
| Thr | O-linked glycosylation |
| Thr | Phosphorylation |
| Trp | Di-oxidation |
| Trp | **Formation of naphthalene** |
| Trp | Mono-oxidation |
| Tyr | **C-amidation (C-terminus)** |
| Tyr | **Phosphorylation** |
| Tyr | Sulfation |
| Val | N-acetylation (N-terminus) |

The mAAs used in this work are highlighted in bold.

often material-intensive and time-consuming. State-of-the-art solubility measurements such as PEG solubility assays, require substantial amounts of material, and have a throughput typically unsuitable for the screening of thousands of candidates[66–69]. Therefore, developing computational methods to predict the intrinsic solubility and aggregation propensity of peptides and proteins with mAAs would be highly beneficial. Laborious solubility measurements could be avoided or greatly reduced by incorporating fast and inexpensive in silico screenings in development pipelines. Although there are

several accurate protein and peptide solubility predictors available as well as predictors for individual amino acids, to our knowledge no sequence-based method can readily handle non-natural amino acids[70–74].

To bridge this gap, here we exploited the CamSol framework for the prediction of intrinsic solubility[75–77] to develop the CamSol-PTM method, which can handle peptides containing mAAs that are of similar size to canonical amino acids. CamSol-PTM is capable of assessing the effect of any kind of small-size noncanonical amino acid on the intrinsic solubility of peptides in aqueous solution at room temperature by combining a range of different physico-chemical property predictors. The absolute solubility of a peptide is the combination of its intrinsic solubility and external factors that impact its solubility such as solvents, ionic strength and pH. By focusing on predicting intrinsic solubility, we aim at creating a general model that can be extended to take external factors into account[77]. The base model is focusing on the intrinsic solubility in aqueous solutions at room temperature. We experimentally validate this approach on variants of three peptides incorporating different mAAs at most positions. The wild-type peptides, which we include in the validation, are glucagon-like peptide-1 (GLP-1), tyrosine tyrosine (PYY), and 18 A.

GLP-1 is a peptide used to treat several disorders, most notably obesity and type-2 diabetes[78–80]. It reduces appetite, glucagon secretion and slows down gastric emptying[80], and has a low risk of inducing hypoglycemia, a common side effect for diabetes drugs[78]. GLP-1 is a 36 amino acid long peptide that when cleaved at the N-terminus produces its active form: GLP-1$_{7-36}$ amide[78]. The drawback of GLP-1 in its native form is that, like most peptides, it has a short half-life and fast clearance rate[80]. The GLP-1 derivatives liraglutatide and semaglutide were developed to overcome this issue[80,81]. The half-life of these drugs is significantly extended compared to its native form by introducing long fatty acid chains that improves drug half-life primarily by enabling albumin binding[82–87].

PYY acts similarly to GLP-1 and is sometimes administered in combination with it to treat obesity, as it is co-released by the body when nutrients are detected[81]. In addition to appetite regulation, it affects energy and glucose homeostasis[81,88,89]. PYY is a gut hormone with a length of 36 amino acids, although its major form is truncated at the N-terminus to give PYY$_{3-36}$[88]. Other truncated variants such as 1-34 and 3-34 are also present but appear to be inactive[81]. The C-terminus of PYY binds four different receptors of the neuropeptide Y receptor family[81,89]. It has a similarly short half-life as GLP-1, approximately 10 minutes[81].

18 A is a derivative of apolipoprotein A (ApoA-1) which is the major component of high-density lipoproteins (HDLs)[2]. Apolipoproteins are complexes that contain lipids and proteins, which transport lipids and other hydrophobic molecules through the body[90]. HDLs can remove cholesterol by decreasing low-density lipoproteins (LDLs) and therefore act against lipid imbalance which is a major cause for cardiovascular diseases[2]. ApoA-1 is a 243 amino acid-long protein that consists of 10 amphipathic α-helices which interact with lipids[2]. 18 A is an 18 amino acid long peptide[91] that mimics these α-helices[2]. Since the original 18 A design, many improvements were made to increase its affinity to lipids and homology to ApoA-1 such as acetylating the N-terminus and amidating the C-terminus[2,90].

For each of these peptides, we screened computationally over 10,000 variants containing combinations of 5 different mAAs. For validation, we then synthesised 30 of those peptides and measured their solubility for the initial set. A second set of 7 peptides containing 4 new mAAs was used to confirm the generalisability of our approach. Our results show that CamSol-PTM can reliably predict the intrinsic solubility of peptides containing mAAs, showing high correlation between predicted and experimentally measured relative solubility.

## Results

### Computational predictions

In this work we exploited the CamSol framework for the accurate prediction of the intrinsic solubility of proteins[75–77] to introduce a method able to predict the effect of mAAs on the solubility of peptides. The original CamSol method predicts the intrinsic solubility of proteins by combining tabulated values of hydrophobicity, charge, and α-helical and β-sheet propensities of the 20 standard amino acids. To extend these tables to a range of different mAAs, information on the physicochemical properties of these mAAs is required (Fig. 1). Because our goal is to estimate the intrinsic solubility of mAA-containing peptides without the need to carry out extensive experimental studies, we build a pipeline in which the physicochemical properties of the mAAs are predicted computationally.

### pKa values

We calculated pKa values of modified side-chains using the recently developed pIChemiSt suite which calculates ionisation constants using pKaMatcher[92]. pKaMatcher matches SMARTS patterns of the mAAs with a list of SMARTS patterns with known pKas[92].

### Hydrophobicity

CamSol uses hydrophilicity values closely related to the inverse of experimental logP values[75]. Here, to develop a predictor of the hydrophobicity of the mAAs, we used a combination of different hydrophobicity calculators to reduce possible biases. After considering the results of several benchmarks, we selected three hydrophobicity predictors: ALOGPS, XLOGP3 and KOWWIN[93–95]. All these methods are machine learning-based, which train their algorithms on different descriptors. ALOPGS[96,97] is based on creating 75 electrotopological-state (E-state) indices trained on the Physprop database (*Syracuse Research Corporation. Physical/Chemical Property Database (PHYSPROP); SRC Environmental Science Center: Syracuse, NY. (1994)*)[93,98]. XLOGP3 is an atomic-based model[99] that uses 87 atomic groups and two correction factors[93]. KOWWIN is fragment-based, using 150 different fragments and 250 corrections[93,100].

Next, we fitted the hydrophobicity values for the 20 natural amino acids as calculated with these predictors to the tabulated CamSol hydrophilicity values. This fit accomplishes two goals. First, the original tabulated values of the 20 natural amino acids do not have to be changed. Second, aligning mAA hydrophilicity values to the original value range bypasses the need to re-fit the parameters used to combine the different biophysical properties in the CamSol framework[75]. We thus calculated the correlation of each of these individual predictors with the original hydrophilicity values of CamSol for the 20 standard amino acids (Supplementary Fig. 1a–c). Using a linear regression analysis, we obtained a fit function to the target values, which showed a higher correlation than with the individual predictors with a Pearson's coefficient of correlation of 0.9 (Supplementary Fig. 1d). Although the combination of the three predictors was accurate, KOWWIN was not suited for the automation of the whole process. Since KOWWIN is only available as part of the EPA suite which only runs on Windows and is not open source, it would be very laborious to include this in the process[101]. However, we found that the accuracy of CamSol-PTM is not significantly affected when using only the other two predictors (Pearson's coefficient of correlation = 0.88) (Supplementary Fig. 1e).

### Secondary structure propensity

We set out to develop a predictor of secondary structure propensity for mAAs based on physico-chemical properties. The values for the 20 standard amino acids are calculated using statistics from the PDB[75]. However, many types of mAAs are either too rare or altogether absent in the PDB, meaning that a new approach was needed. We considered the following characteristics: molecular weight (MW), number of

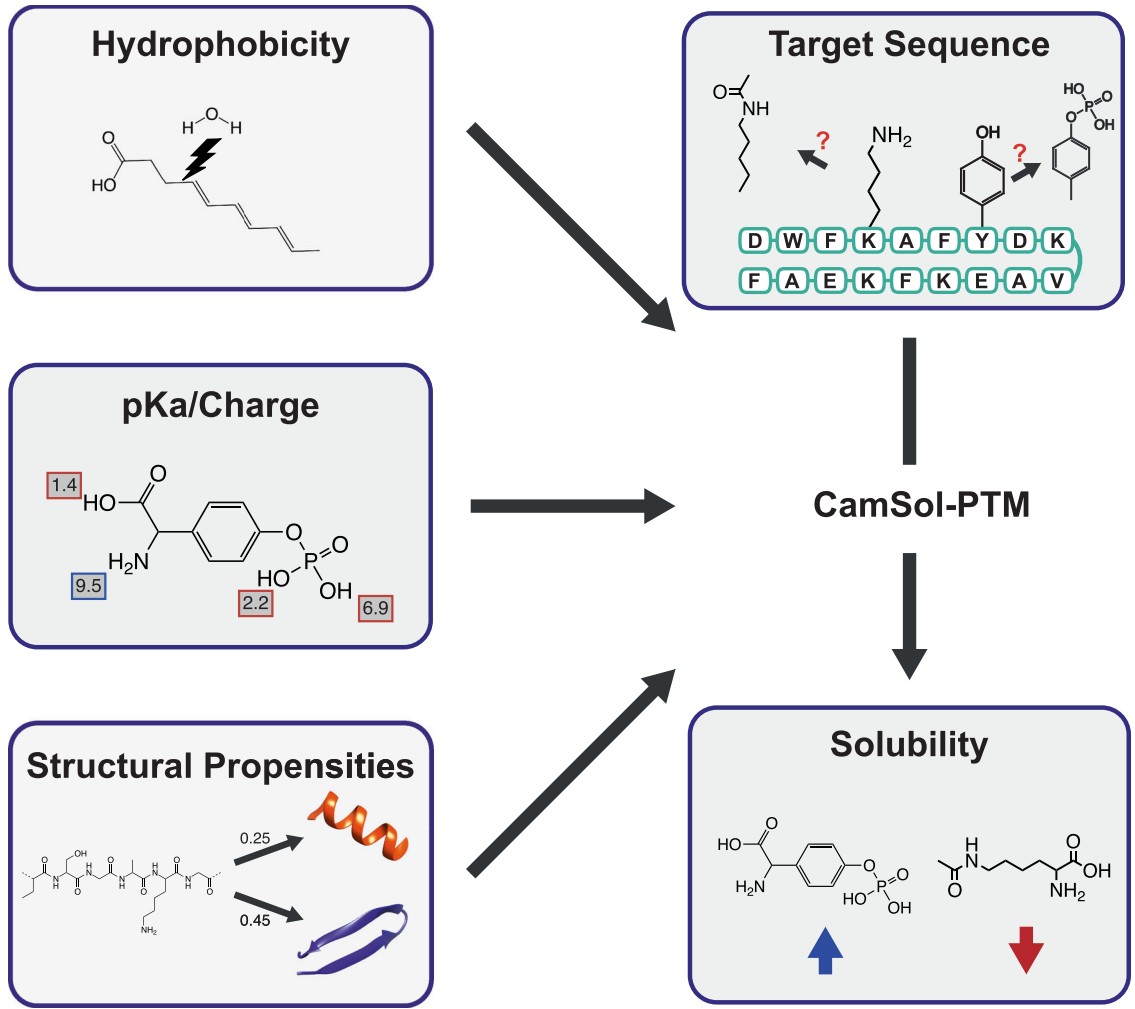

**Fig. 1 | Workflow for optimising the solubility of peptides containing modified amino acids (mAAs) using CamSol-PTM.** A linear combination of ALOGPS[96,97] and XLOGP3[100] is employed to determine the hydrophobicity values. pIChemiSt suite[92] is used to predict the pKa values of mAAs. Structural propensities are calculated using a separate predictor that gives an estimate on the likelihood of finding a mAA in an α-helix or a β-sheet. The predictor employs a combination of the number of hydrogen donors and acceptors, the number of rotational bonds, molecular weight and the topological polar surface area. All this information is fed into the CamSol-PTM algorithm to predict the effect of mAAs on the solubility of a peptide.

hydrogen donors ($H_D$) number of hydrogen acceptors ($H_A$) number of rotational bonds (RB) and topological polar surface area (TPSA). The information on these properties for all standard amino acids and the mAAs used in this work were initially gathered from https://pubchem.ncbi.nlm.nih.gov/. The final version of CamSol calculates these values using the python module RDKit. To determine which combination of properties would yield the best predictor, we explored a series of linear equations for different combinations of these five properties, such for example

$$p_i^\alpha = \alpha_{MW} * MW_i + \alpha_{TPSA} * TPSA_i + \alpha_{RB} * RB_i, \tag{1}$$

where $p_i^\alpha$ is the calculated α-helical propensity of amino acid $i$ and $\alpha_X$ are the linear coefficients to be fitted. For each combination of the properties, we fitted a function to the tabulated secondary structure propensity values of the standard amino acids. We excluded glycine and proline, since these two amino acids have unusual secondary structure propensity values and would skew the fit. Moreover, we also used the resulting secondary structure propensity values of each of these combinations within the CamSol-PTM framework to predict the solubilities of all peptides. To choose which secondary structure propensity predictor was the most promising we looked at the Pearson's coefficients of correlation between the predicted secondary

structure propensity values and their tabulated counterparts as well as at the correlation between the experimental and predicted solubility data for the 30 peptide variants. The choice of propensities that offered the best combination of high correlation for the secondary structure propensities as well as the high correlation between the predicted and experimental solubilities while simultaneously using as few parameters as possible was $H_D$ and TPSA for α-helical propensities ($R = 0.59$) and MW, RB and TPSA for β-sheet propensities ($R = 0.69$, Supplementary Fig. 2).

**Sequence parser**

As a 1-letter alphabet is not available for all possible mAAs, we parsed the input sequence as follows. mAAs are added to the standard protein sequence as a three-letter code in square brackets (e.g. Ala-norleucine-Gly would be denoted as 'A[NLE]G'). A careful literature research regarding nomenclature for denoting mAAs showed that there is currently no widely used and simultaneously easy-to-read format for coding mAAs. Therefore, we kept the implementation flexible in order for any kind of nomenclature to be used.

**Choice of modifications**

To decide the set of mAAs for an initial testing, we considered a range of different functionalities. Acetylation of native lysine (NAC) residue is

a common PTM with great impact on the properties of a peptide, as it removes a positive charge. Aminoisobutyric acid (AIB) is often used to make peptides more resistant against peptidases as it is not easily recognised[79]. Norleucine (NLE) is closely related to the natural amino acids leucine, valine and isoleucine, but with its longer non-branched aliphatic chain offers a slightly different functional group; it is also typically used as a non-oxidation labile methionine substitution. Cyclohexylalanine (CHA) offers a unique functionality due to its highly hydrophobic non-aromatic six-membered ring. Citrulline (CIT) offers alternative functionality that resembles arginine. Moreover, we also implemented modifications to the N- and C-termini of peptide scaffolds: N-acetylated aspartic acid, C-amidated phenylalanine and C-amidated tyrosine as these were already included in the base peptides. With this mix of new functionalities and some closely related mAAs we aimed to cover a broad chemical space.

### Peptide design

Due to the limit of the number of possible variants that could be synthesised and purified in this study, we wanted to ensure that our designs covered the largest possible chemical space while exploring a broad range of solubility values. For each peptide we designed five variants each containing one mAA. We chose alanine residues as the starting point for single modifications to have a common baseline for all mAAs. Additionally, we screened all possible combinations of double modifications for each peptide. The first step, however, was to define regions for each peptide that allowed for modification without interfering with the binding capabilities and specific folds.

GLP-1 consists of two α-helices separated by a linker. We chose the first alanine in the linker region (residue 24) as the starting point for single-site modifications. For the double-site modifications, we further excluded the following residues due to their essential role in binding: 7His, 8Ala, 9Glu, 11Thr, 12Phe, 13Thr, 14Ser, 16Val, 17Ser, 18Ser, 19Tyr, 20Leu, 21Glu, 26Lys, 28Phe, 29Ile, 31Tyr, 32Leu, 33Val, 34Lys.

PYY consists of a proline-rich α-helix at the N-terminus which forms H-bonds with the α-helix that comprises the rest of the molecule. Hence, we chose an alanine in the proline-rich region to perform the single-site modifications. For the double-site modifications, we excluded all prolines and hydrogen-bonding residues, i.e. R, H, K, D, E, N, Q.

18 A has an amphipathic nature that is convenient to maintain. Therefore, for the single-site modifications, we chose alanine at position 10, located on the edge between the two sides. For the double-site modifications, we ensured that the hydrophilic residues (D, E, K) were only replaced with hydrophilic modifications (CIT, AIB) and hydrophobic residues (W, F, A, V) were only replaced with hydrophobic mAAs (CHA, NAC, NLE).

Given these constraints, we screened over 50,000 mAA variants using CamSol-PTM. From all these possible variants for double modifications, we chose at least one variant where one of the modifications is rather small, e.g., L to NLE, F to CHA, A to AIB or R to CIT. For the remaining three doubly modified variants per peptide, we chose one variant each predicted as either very soluble, very insoluble or average in solubility. The sequences of the designed peptides are given in Table 2.

### Generation of experimental data

Relative solubility was measured using a recently developed PEG precipitation assay[66]. For all PYY variants the standard assay worked well, and no changes had to be implemented (Fig. 2a). Variants 27 and 28 were completely soluble whereas variant 30 was already insoluble in the absence of PEG, and variant 29 proved to be difficult to produce and purify. Therefore, these four are not reported in Fig. 2. 18 A and its variants proved more complicated, as most variants were completely soluble up to 30% PEG. We therefore switched from PEG to ammonium sulphate (AMS) precipitation (Fig. 2b), as it has been shown that relative solubility measurements with PEG and AMS are correlated[102].

Moreover, to ensure that the results stemming from the AMS assay are consistent and reliable, we performed the 18 A experiments twice independently on different days. The results confirmed that they are indeed replicable, and we were therefore confident to use them for the validation of our approach (Supplementary Fig. 3). Two variants, namely variant 17 and 18 proved to be completely insoluble and variant 12 was not produced in sufficient amounts. Therefore, these are not reported in the figures. The last set of variants stemming from GLP-1 had the inverse problem, as most variants proved to be very insoluble. Even at final concentrations of 0.33 mg/mL (instead of 1 mg/mL) most variants remained insoluble. We used ultracentrifugation to determine the relative solubilities of the GLP-1 variants (Table 3). To confirm the reliability of this method we replicated the results on a different day with the same stock solutions (Supplementary Fig. 4).

### Correlation between predicted and experimental solubility values

By comparing the computational predictions with the experimental data, we found high correlations between the two data sets. The Pearson's coefficients of correlation for the PYY variants are 0.78, 0.81 for the 18 A variants and 0.58 for the GLP1 variants (Fig. 3). To ascertain that these findings were not merely a coincidence, we designed a second set of PYY variants containing four new mAAs and measured their solubilities (Fig. 2c). The results are depicted in Fig. 3a in ochre. Variant 32 is not depicted as it was not possible to measure its solubility with the PEG Assay. The overall Person's coefficient of correlation for the combined set of PYY variants is 0.6.

Encouraged by the results of the experimental validation, we set out to generalise the computational approach to broaden its applicability to more mAA types. We set up a web server under https://www-cohsoftware.ch.cam.ac.uk/index.php/camsolptm for academic user to freely use our method. We automated the process of adding new mAAs by replacing the hydrophobicity predictor with the Crippen tool from RDKit. If a user would like to predict the solubility of a peptide containing a noncanonical amino acid that has not been implemented yet, only the SMILES code is required. By providing this information, the web server will automatically calculate the necessary properties for this mAA in order for the user to include it in the prediction.

To demonstrate the speed of the automation, we incorporated the whole set of non-canonical amino acids that Amarasinghe et al. recently produced through extensive in silico screenings[65]. CamSol-PTM can calculate about 15 new residues per second on a single CPU core. We then designed 40,000 single mutational variants of a 60 residue-long Nrf2 peptide fragment centred around the mutational sites Leu76, Asp77, Glu78 and Leu84, which were previously identified[65]. We predicted the intrinsic solubility for each of these variants which took 8 min on a single CPU core (around 80/s) and plotted the distribution of the solubilities (Fig. 4). By analysing the tail ends of the distribution, we found that, in agreement with chemical intuition, mAAs that contain many hydrogen bonding residues such as those containing nitrogen and oxygen atoms are among the most solubility-promoting residues (Supplementary Fig. 5). The mAAs that most negatively affected the solubility largely contain several aromatic rings and often halogens such as chlorine or bromine (Supplementary Fig. 6).

## Discussion

Peptide intrinsic solubility is one of the most crucial parameters that determine the likelihood of a peptide to be successfully developed into a commercial drug product. Application of automated, predictive technologies with high throughput and low compound requirements are very useful for efficient early profiling and optimization of physicochemical properties, such as solubility during early discovery program allowing for more comprehensive screenings and faster development times.

**Table 2 | List of peptides designed to verify the CamSol-PTM predictions**

| Compound | Peptide | Sequence | Modifications |
|---|---|---|---|
| 1 | GLP1 | HAEGTFTSDVSSYLEGQAAKEFIAWLVKGR | None |
| 2 | GLP1 | HAEGTFTSDVSSYLEGQ[CHA]AKEFIAWLVKGR | A -> CHA |
| 3 | GLP1 | HAEGTFTSDVSSYLEGQ[NLE]AKEFIAWLVKGR | A -> NLE |
| 4 | GLP1 | HAEGTFTSDVSSYLEGQ[NAC]AKEFIAWLVKGR | A -> NAC |
| 5 | GLP1 | HAEGTFTSDVSSYLEGQ[AIB]AKEFIAWLVKGR | A -> AIB |
| 6 | GLP1 | HAEGTFTSDVSSYLEGQ[CIT]AKEFIAWLVKGR | A -> CIT |
| 7 | GLP1 | HAEGTFTSDVSSYLEGQ[CHA]AKEFIAWLVKG[CIT] | A -> CHA, R -> CIT |
| 8 | GLP1 | HAE[AIB]TFTSDVSSYLEGQAAKEF[CIT]AWLVKGR | G -> AIB, I -> CIT |
| 9 | GLP1 | HAEGTFTS[CHA]VSSYLEGQAAK[NAC]FIAWLVKGR | D -> CHA, E -> NAC |
| 10 | GLP1 | HAE[NLE]TFTSDVSSYLEG[CIT]AAKEFIAWLVKGR | G -> NLE, Q -> CIT |
| 11 | 18 A | [ntDAC]WFKAFYDKVAEKFKEA[ctFAD] | None |
| 12 | 18 A | [ntDAC]WFKAFYDKV[CHA]EKFKEA[ctFAD] | A -> CHA |
| 13 | 18 A | [ntDAC]WFKAFYDKV[NLE]EKFKEA[ctFAD] | A -> NLE |
| 14 | 18 A | [ntDAC]WFKAFYDKV[NAC]EKFKEA[ctFAD] | A -> NAC |
| 15 | 18 A | [ntDAC]WFKAFYDKV[AIB]EKFKEA[ctFAD] | A -> AIB |
| 16 | 18 A | [ntDAC]WFKAFYDKV[CIT]EKFKEA[ctFAD] | A -> CIT |
| 17 | 18 A | [ntDAC]W[CHA]KAFYDKV[CHA]EKFKEA[ctFAD] | F -> CHA, A -> CHA |
| 18 | 18 A | [ntDAC]WFK[CHA]FYDKVAEKFKE[NLE][ctFAD] | A -> CHA, A -> NLE |
| 19 | 18 A | [ntDAC]WF[AIB]AFYDKVAEK[CHA]KEA[ctFAD] | K -> AIB, F -> CHA |
| 20 | 18 A | [ntDAC]W[NAC]KAFYDKVAEK[NLE]KEA[ctFAD] | F -> NAC, F -> NLE |
| 21 | PYY3-36 | IKPEAPREDASPEELNRYYASLRHYLNLVTRQR[ctYAD] | None |
| 22 | PYY3-36 | IKPEAPRED[CHA]SPEELNRYYASLRHYLNLVTRQR[ctYAD] | A -> CHA |
| 23 | PYY3-36 | IKPEAPRED[NLE]SPEELNRYYASLRHYLNLVTRQR[ctYAD] | A -> NLE |
| 24 | PYY3-36 | IKPEAPRED[NAC]SPEELNRYYASLRHYLNLVTRQR[ctYAD] | A -> NAC |
| 25 | PYY3-36 | IKPEAPRED[AIB]SPEELNRYYASLRHYLNLVTRQR[ctYAD] | A -> AIB |
| 26 | PYY3-36 | IKPEAPRED[CIT]SPEELNRYYASLRHYLNLVTRQR[ctYAD] | A -> CIT |
| 27 | PYY3-36 | IKPEAPRED[CIT]SPEELNRYYASLRHY[NLE]NLVTRQR[ctYAD] | A -> CIT, L -> NLE |
| 28 | PYY3-36 | IKPEAPREDA[NLE]PEELNRYYA[NLE]LRHYLNLVTRQR[ctYAD] | S -> NLE, S -> NLE |
| 29 | PYY3-36 | IKPE[AIB]PREDASPEELNRYYA[NAC]LRHYLNLVTRQR[ctYAD] | A -> AIB, S -> NAC |
| 30 | PYY3-36 | [AIB]KPEAPREDASPEELNRYYASLRHYLNL[AIB]TRQR[ctYAD] | I -> AIB, V -> AIB |
| 31 | PYY3-36 | IKPEAPRED[DAP]SPEELNRYYASLRHYLNLVTRQR[ctYAD] | A -> DAP |
| 32 | PYY3-36 | IKPEAPRED[NAP]SPEELNRYYASLRHYLNLVTRQR[ctYAD] | A -> NAP |
| 33 | PYY3-36 | IKPEAPRED[TBA]SPEELNRYYASLRHYLNLVTRQR[ctYAD] | A -> TBA |
| 34 | PYY3-36 | IKPEAPRED[OPO]SPEELNRYYASLRHYLNLVTRQR[ctYAD] | A -> OPO |
| 35 | PYY3-36 | IKPE[CHA]PREDASPEELNRYYASLRH[OPO]LNLVTRQR[ctYAD] | A -> CHA, Y -> OPO |
| 36 | PYY3-36 | IKPE[OPO]PREDASPEELNRYYASLRHYLN[TBA]VTRQR[ctYAD] | A -> OPO, L -> TBA |
| 37 | PYY3-36 | [CIT]KPEAPREDASPEE[AIB]NRYYASLRHY[DAP]NLVTRQR[ctYAD] | I -> CIT, L -> AIB, L -> DAP |

Initially, for each peptide, nine variants were designed. Five include single-site modifications, one is a double-site modification where one modification is small and three are random double-site modifications. In a second step another seven variants for PYY were designed (31–37) containing four new mAAs.

Non-canonical amino acids are often used to introduce unique functionalities to drugs such as peptidase resistances[1,4,17,19,31–36] or increase binding affinities[4,19]. However, experimental methods to evaluate the developability of peptides containing mAAs are typically costly, and current computational approaches lack the capability of capturing the effects of mAAs on the solubility of peptides. To address this problem, we have presented CamSol-PTM, a software that predicts the intrinsic solubility in aqueous solution at room temperature of peptides and proteins containing non-canonical amino acids based on the physicochemical properties of their amino acid sequences[75–77].

To test the CamSol-PTM predictions, 30 variants of 3 peptides containing 5 different mAAs were chosen from a preliminary screen of over 50,000 designs. The peptides were produced and purified, and their solubilities were experimentally measured. The comparison between measurements and predictions showed that CamSol-PTM can predict the intrinsic solubility of peptides and proteins containing

mAAs with high accuracy (Pearson's coefficients of correlation 0.72 on average).

We confirmed the generalisability of our approach by designing a second set of PYY variants with four new mAAs and measured their solubility and compared it to our predictions. The high overall Pearson's coefficient of correlation for the whole set of PYY variants – although being slightly lower at 0.6 - showcases the robust applicability of our method.

Although the wild types of the peptides tested in this study tend to form α-helices, we do not expect our method to be significantly biased towards this type of secondary structure. First, most parameters, including the ones to calculate the solubility score for individual amino acids and the parameters used to determine the overall solubility of a protein are identical to original CamSol method which was trained on a wide range of varying secondary structure. Second, the mAAs tested were not merely α-helical promoting residues and are therefore not biased towards α-helical structures.

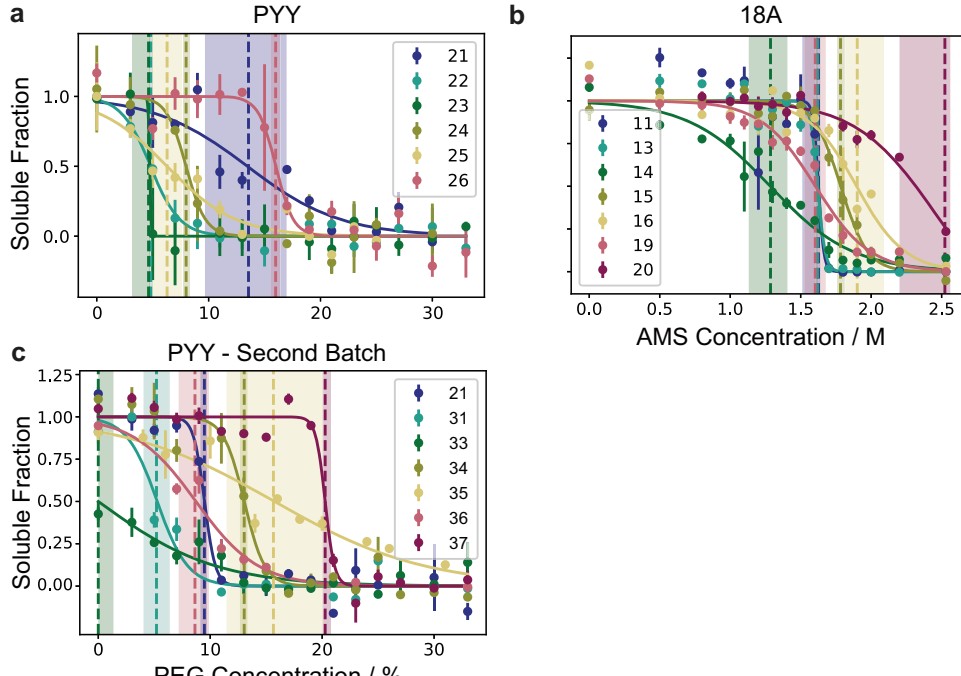

**Fig. 2 | Experimental solubility data for peptides generated using the PEG solubility assay.** Solubility curves determined using a recently developed PEG solubility assay[66] for all successfully synthesised variants (all designs except variants 12 and 29) that are neither completely soluble (variants 27 and 28) nor insoluble (variants 17, 18 and 30) for: PYY (**a**), 18 A (**b**) and the second batch of PYY variants (**c**). For 18 A AMS was used instead of PEG. $PEG_{1/2}/AMS_{1/2}$ values are shown as a vertical line with the shaded region depicting the 95% confidence interval. PEG percentages are mass/volume[66]. Error bars represent the standard error of the experimental measurements across technical replicates ($n = 4$ for PYY and PYY – Second Batch, $n = 2$ for 18 A) where the centre represents the mean. Source data are provided as a Source Data file.

It has been recently shown that by creating new unnatural amino acids in silico, it is possible to create effective new compounds, thus demonstrating the potential of incorporating more diverse mAAs into the drug development process[65]. By automating the process of adding new mAAs to CamSol-PTM, the method is now capable of predicting the effects of small mAAs on the solubility of proteins and peptides. We have demonstrated the speed and versatility of the method by adding all 10,000 mAAs reported recently by Amarasinghe et al. to our method and predicting the solubility of 40,000 mutational variants of a Nrf2 peptide fragment[65].

We acknowledge that although our method increases the chemical space that can be covered by solubility predictions by several orders of magnitude compared to the 20 natural amino acids, it is currently restricted to modifications that are of similar size to canonical amino acids. Further developments will be required to assess the effects of larger modifications such as lipids or glycans on the intrinsic solubility of peptides.

We envisage that the CamSol-PTM method will substantially aid in the understanding of the effects of non-canonical amino acids on the intrinsic solubility of proteins and peptides. As with previous versions, it can also be used to identify aggregation hot spots by analysing the solubility profiles. Moreover, we except it to be a valuable tool for drug development as it enables the fast and accurate solubility prediction of peptides containing modified amino acids.

## Methods

### Materials
N-α-D-Fmoc protected amino acids were sourced from Bachem AG (Switzerland). Synthesis reagents and solvents were all obtained from NovaBioChem, Merck (UK) and used without further purification. Peptide sequences were prepared using automated microwave-assisted solid phase peptide synthesis using the CEM Liberty Blue synthesiser and Fmoc chemistry with standard side chain protecting groups.

### Peptide synthesis
All peptides were synthesised as C-terminal carboxamides on Rink Amide MBHA resin (loading 0.23 mmol/g, 100–200 mesh) on a 0.1 mmol scale using DIC/HOBt activation. All amino acids were double coupled for 4 min at 75 °C, with the instrument set to deliver the N-α-Fmoc-amino acid solutions (0.2 M solution in DMF), HOBt (1.0 M solution in DMF) and DIC (1.0 M solution in DMF). Deprotection cycles were performed using 20% piperidine solution (in DMF, + 0.1 mol HOBt) for 1 min at 90 °C following each cycle. Crude peptides were cleaved from the resin using a cleavage cocktail containing TFA (95%), triisopropylsilane (2.5%) and water (2.5%) for 4 hours at room temperature. The resin was removed by filtration and the cleavage solution removed *in vacuo*. The peptides were precipitated by addition of diethyl ether, isolated by centrifuge at 3500 rpm and dried under a flow of dry nitrogen.

**Table 3 | Experimental solubility data for the GLP-1 variants generated using ultracentrifugation**

| Variant | 1 | 2 | 3 | 4 | 5 | 6 | 7 | 8 | 9 | 10 |
|---|---|---|---|---|---|---|---|---|---|---|
| Run 1 / mg/mL | 0.58 | 0 | 0.09 | S | 0.09 | S | 0.84 | S | 0 | 0.15 |
| Run 2 / mg/mL | 1.38 | 0 | 0.16 | S | 0.07 | S | 0.82 | S | 0 | 0.12 |

Results of two independent ultracentrifugation runs measuring the solubility of the GLP-1 variants. S symbolizes the outcomes in which no precipitation occurred.

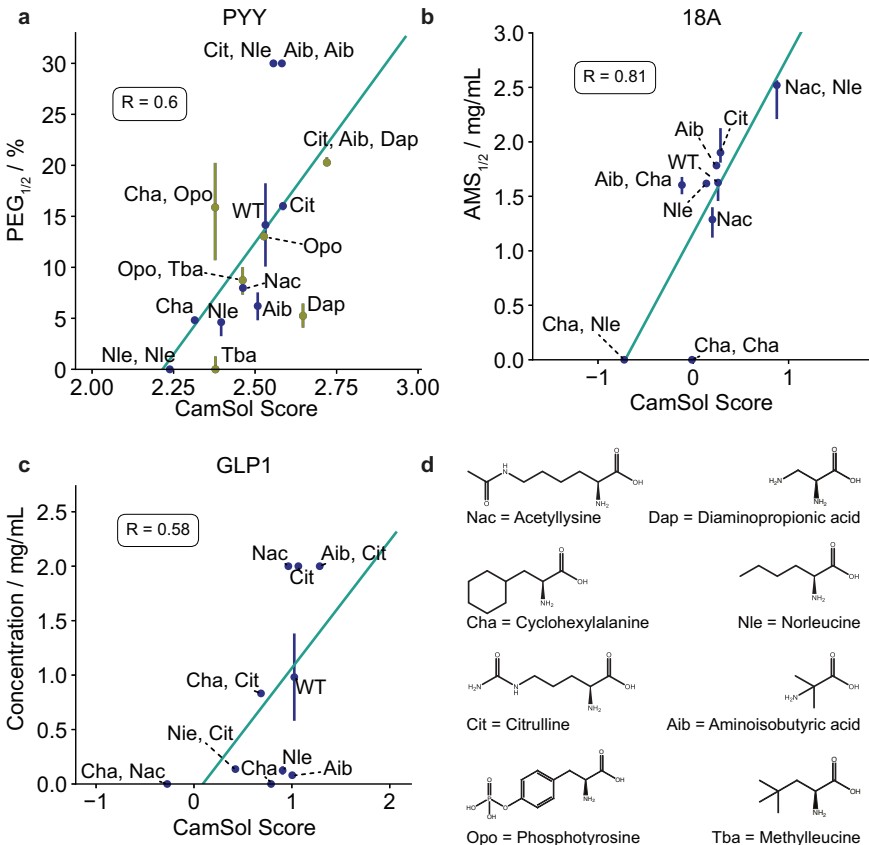

**Fig. 3 | Correlation between experimental and predicted solubility values of the designed peptides containing mAAs.** The Pearson's coefficients of correlation are 0.6 for PYY (0.78 for the initial set) (**a**), 0.81 for 18 A (**b**) and 0.58 for GLP1 (**c**). mAAs that were used are shown in (**d**). The two designs (12 and 29) that could not be produced in sufficient amounts were removed from the analysis. Error bars in a and b represent the 95% confident intervals of the $PEG_{1/2}$ values stemming from the sigmoidal function fitted through the experimental measurements shown in Fig. 2 (technical replicates $n = 4$ for a and $n = 2$ for **b**) where the centre represents the mean. Error bars in c represent the standard error of the experimental measurement shown in Table 3 across technical replicates ($n = 2$) where the centre represents the mean. Source data are provided as a Source Data file.

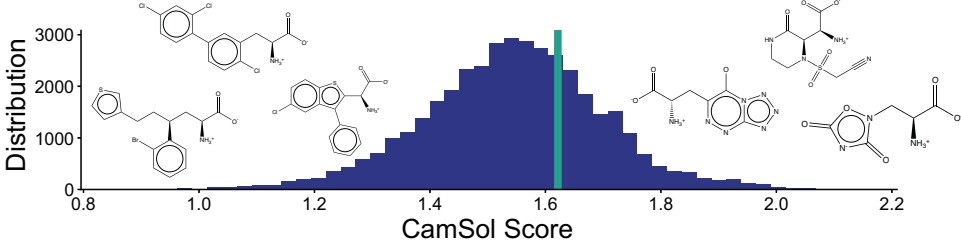

**Fig. 4 | Solubility distribution of 40,000 variants of the Nrf2 peptide fragment.** Single mutants were designed containing one of the recently reported 10,000 mAAs[65] at one of four positions (Leu76, Asp77, Glu78, Leu84). Solubility of the wild-type peptide is highlighted with a turquoise line. Analysis of the tail ends of the distribution revealed that mAAs that contain many hydrogen-bonding promoting atoms such as nitrogen and oxygen are predominantly found in the highly soluble region, whereas mAAs with halogens such as chlorine and bromine and aromatic rings are mostly found in the insoluble region. The vertical line depicts the CamSol score for the wild type Nrf2 peptide fragment. Source data are provided as a Source Data file.

## Peptide purification and analysis

Prior to purification, crude peptides were reconstituted in 5% acetonitrile in water (v/v) or dissolved in TFA and diluted with ACN/Water/TFA 50/50/0.1 mixture and filtered (0.4 μm, PTFE). The purifications were performed by preparative HPLC (Waters Fraction Lynx system connected to a PDA detector and Waters SQD mass spectrometer) using a Waters Atlantis T3 OBD column, Waters XSelect CSH Fluoro Phenyl OBD column or a Waters XBridge C18 OBD column with a focused acetonitrile gradient at room temperature. The mobile phases used were either at acidic or neutral conditions. For specific conditions see Supplementary Data 1. Fraction collection was triggered on either a UV threshold or target mass intensity threshold, the UV trace was monitored at 230 nm. The collected fractions were pooled and analysed on a C8 or a C18 column by Waters UPLC system (or Agilent 1200 series gradient HPLC system) using a linear acetonitrile gradient at acidic conditions (Supplementary Data 1). UV purity was estimated to between 82 and 99% at 210 nm or 230 nm on a Waters H-Class UPLC system with a PDA, Waters SQD mass spectrometer (or Waters 3100 system). Target masses were verified against theoretical values on the mass spectrometer operating in ES+ mode.

## Solubility assay

Aliquots of 1 mg were prepared from the purified and lyophilised stocks. The solubility of the PYY and 18 A variants was measured using the PEG solubility assay that was developed in this group[66]. Briefly, a precipitant is titrated in increasing concentration to a fixed concentration of protein to induce precipitation of the protein. The samples are incubated for 48 h at 4° after mixing. The samples are centrifuged and the remaining protein concentration is measured in the supernatant using a plate reader. PYY and 18 A variants were dissolved in 10 mM citrate 10 mM phosphate buffer at pH 7 for a final concentration of 3 mg/mL. The assay was run with 50% 6000 PEG for PYY and with 3.8 M AMS for 18 A. To improve throughput, a multichannel robot was employed to measure several peptides at once with the workflow being kept the same as described previously[66]. The solubility of the GLP1 variants was measured with ultracentrifugation as follows: The peptides were dissolved in 10 mM citrate 10 mM phosphate buffer at pH 7 for a final concentration of 2 mg/mL. 120 μL of each sample were centrifuged using an OptimaTLX Ultracentrifuge and spinning for 30 min at 500,000 g at 4 °C. The supernatant was removed, and the peptide concentration was measured using a NanoDrop.

## Reporting summary

Further information on research design is available in the Nature Portfolio Reporting Summary linked to this article.

## Data availability

All peptide sequences are given in Table 2 and Supplementary Data 1. All data necessary to replicate, evaluate or extend the research presented in this article are provided throughout the article, the supporting information and the Source Data file. All predicted values are provided in the Source Data file and can be replicated by using the webserver under https://www-cohsoftware.ch.cam.ac.uk/index.php/camsolptm. Information on peptide production and purification are included in the supporting information. Source data are provided with this paper.

## Code availability

This method is available as a web server which is free for academic users after registration at https://www-cohsoftware.ch.cam.ac.uk/index.php/camsolptm. For industry users it is possible to purchase a license for the CamSol method from Cambridge Enterprise.

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

## Acknowledgements

M.O. is a PhD student funded by AstraZeneca. P.S. is a Royal Society University Research Fellow (URF\R1\201461). The project was supported by the Wellcome Trust (203249/Z/16/Z).

## Author contributions

M.O. and R.J.D.K. performed experiments and M.O. carried out data analysis. H.L.B., A.N., P.Z. and W.S. synthesize the peptide variants and H.L.B. and A.L.W purified and analysed them. M.O. and P.S. wrote the software. M.O., P.S and M.V. wrote the original draft of the manuscript. H.L.B., A.L.G.d.S., L.D.M, W.S., A.L.W. and W.C. edited the manuscript. M.O., P.S., A.L.G.d.S., W.C., L.D.M. and M.V. conceived and A.L.G.d.S., L.D.M., P.S. and M.V. supervised the project.

## Competing interests

The authors declare the following competing financial interest(s): H.L.B., A.L.G.d.S., A.L., A.N., P.Z., W.S., L.D.M and W.C. are employees of AstraZeneca and may own stocks or stock options. The remaining authors declare no competing interests.
