## [Peer Review File · Nature Communications]

REVIEWER COMMENTS

Reviewer #1 (Remarks to the Author):

In the manuscript "Sequence-based prediction of the solubility of peptides containing non-natural amino acids" by Oeller et al the authors seek a computational method to provide a fast, accurate method for predicting the solubility of a variety of modified and non-natural amino acid residues. They combined some standard physical-chemical properties with some existing fitted measures and performed a standard linear regression to fit. The authors also performed a standard rapid PEG relative solubility test for a small selected set of experimental verification examples. The comparison of the computational and experimental results had modest pearson correlations roughly between .6 and .8 which may be useful in some high through put screens but is not impressive for yielding new insights into solubility. Little analysis of the reason for outliers (or successes) was given.

Reviewer #2 (Remarks to the Author):

The manuscript NCOMMS-23-09868 presents solubility prediction and validation of mAAs. The usage of mAAs is of special interest in the development of peptide-based drugs, and thus, the study is meaningful. The manuscript is well written and describes the approach for solubility prediction to reduce the experimental effort for laborious solubility experiments for drug development. However, the manuscript suffers several important aspects which I expected. Therefore, I would be delighted to accept this manuscript after such changes have been taken into account.

General remarks:

Figure S5 and S6: Please clarify which chemical structure is which mAAs. These two figures contain important information, but they are poorly presented.

Figures in general: Please be consistent with the presentation of your figures. For me, Figure 3 looks like it was copy-pasted. Please use a graphical tool to present homogeneous figures.

Figure 1: Structural properties: 0.25 and 0.45 for the different folding of the peptides. What happened to the other 0.3? Is it percent? Please clarify.

Figure 2c: this should be a table rather than a figure.

Solubility of one compound depends on several things: the medium (solvent, ionic strength, pH...), temperature, and solid form. This has been totally ignored by the authors. Please clarify in the abstract and everywhere else where it appears to be relevant, even in the title I'd expect "Aqueous solubility at x°C".

Abstract:

- What is the abbreviation "CamSOI-PTM" standing for? Please clarify.
- Please define the chemical space (solvent/T, pH).

Intro:

- The authors ignore other available methods to predict solubility. Molecular models that account for the interactions in the liquid phase do exist, which need as input the melting properties of the solids. Such methods are described, e.g., <https://doi.org/10.1039/D0RA08947H>

Paragraph 5:

- Also, here, please specify which chemical space.
- Please name some methods for state-of-the-art solubility measurements

Paragraph 6:

You name the peptides PYY, GLP-1, and 18A at the end of paragraph 6. However, in the next paragraph, you start with GLP-1, followed by PYY. Please rephrase and start with 1) PYY, 2) GLP-1, and 3) 18A.

Results:

- pKa values calculation: Can you please specify how the pICHemiSt suite calculates the pKa value?
- Please enumerate equations
- Equation for p_i^{α} : In the text, the αx should be written in italic letters
- experimental solubility: It is not clear which values are presented in Figure 2. Are these values mass% or volume%? Also, the solubility fraction should have a unit.

Material and Methods:

- Materials:

Can the authors comment on properties such as hydrate or solvate formation of the protected amino acids? This is crucial in any solubility experiment, since the solid form critically influences the solubility that is expected. Usually, this is measured by XRD, which is standard in the community of measuring solubility data. For compounds that change their solid form upon precipitation and being contacted by the solvent, the method the authors presented is not valid.

- Solubility assay:

"Solubility was measured with ultracentrifugation". I think, no. For me, it is not clear how solubility can be measured with ultracentrifugation. Is it an optical determination of the solubility? Please specify and add this information to the manuscript. Information on the media that were used, which water solvent, which ph... has to be added. pKa values might be useful to convert all the solubility data to pH 7, but i guess the pH of the samples has not been measured. It will have an effect on solubility.

Reviewer #3 (Remarks to the Author):

Sequence-based prediction of the solubility of peptides containing non-natural amino acids

In this manuscript, Oeller et al. present a sequence-based method to predict, for the first time, the solubility of peptides containing modified amino acids (mAAs). In short, the authors built a pipeline that predicts the physicochemical properties of any mAAs to incorporate them in the widely used CamSol framework. Solubility predictions for three peptides containing a combination of five different mAAs — for a total of 30 peptide variants — are validated using three different experimental methods.

This is a major update to the CamSol framework. Its novelty, significance and accessibility (through a web server) make it a good fit for the broad audience of Nature Communications, especially those in the fields of biotechnology and drug discovery. The manuscript is clear and does a good job summarizing the current state of the art.

I have, however, a few suggestions to strengthen the work and make it suitable for publication:

Major comments

- Although the method can potentially predict the solubility of peptides containing any mAAs, the experimental validation is performed on mAAs that are of similar size to canonical amino acids. The authors acknowledge this restriction in the discussion, but it should also be mentioned in the abstract and introduction.

- The peptides chosen by the authors are structurally very similar since all three consist mainly of α -helices, thus ignoring the structural space covered by beta-sheets and beta-strands. This class of secondary structure is structurally very different to α -helices and is more often involved in aggregation events. Furthermore, the authors used the experimental solubilities of the 30 peptide variants to choose, a posteriori, the best combination of parameters for the secondary structure propensity predictor. Since all these peptides consist of α -helices, this could lead to the overfitting of this specific class. To claim the generalizability of the method, we ask to include at least one extra peptide in the validation that consists mainly of beta-sheets, such as a beta-hairpin.

Minor comments

- The CamSolPTM web server needs the SMILES codes of mAAs to predict the solubility of peptides containing non-standard amino acids. It will be useful for potential users to provide a table with the SMILES codes of the most common mAAs.

- Besides an overall solubility score assigned to the entire sequence, CamSolPTM yields a solubility profile (one score per residue in the protein sequence) where regions with scores larger than 1 denote highly soluble regions, while scores smaller than -1 poorly soluble ones. Previous studies by this group using the CamSol framework have used these per residue values to identify aggregation hot spots. Is the same true for CamSolPTM?

- In Figure 3, include in the figure description which variants are removed from the analysis.

Reviewer: 1

In the manuscript "Sequence-based prediction of the solubility of peptides containing non-natural amino acids" by Oeller et al the authors seek a computational method to provide a fast, accurate method for predicting the solubility of a variety of modified and non-natural amino acid residues. They combined some standard physical-chemical properties with some existing fitted measures and performed a standard linear regression to fit. The authors also performed a standard rapid PEG relative solubility test for a small selected set of experimental verification examples. The comparison of the computational and experimental results had modest pearson correlations roughly between .6 and .8 which may be useful in some high through put screens but is not impressive for yielding new insights into solubility. Little analysis of the reason for outliers (or successes) was given.

We thank the reviewer for these comments. We would like to emphasise that the novelty of our software is the assessment of non-natural amino acids. To our knowledge, this is the first sequence-based method that can predict the solubility of peptides that contain non-natural amino acids. We demonstrate that our software can facilitate the screening of possible mutations, increasing throughput by four orders of magnitude. We have now amended the abstract and discussion section to explain more effectively these points.

Reviewer: 2

The manuscript NCOMMS-23-09868 presents solubility prediction and validation of mAAs. The usage of mAAs is of special interest in the development of peptide-based drugs, and thus, the study is meaningful. The manuscript is well written and describes the approach for solubility prediction to reduce the experimental effort for laborious solubility experiments for drug development. However, the manuscript suffers several important aspects which I expected. Therefore, I would be delighted to accept this manuscript after such changes have been taken into account.

We appreciate the reviewer's constructive comments. Incorporating the suggested revisions has improved the manuscript.

General remarks:

Figure S5 and S6: Please clarify which chemical structure is which mAAs. These two figures contain important information, but they are poorly presented.

We adjusted the figures accordingly. Now each structure is accompanied by its unique three-character identifier. We also decluttered the figures to make them clearer.

Figures in general: Please be consistent with the presentation of your figures. For me, Figure 3 looks like it was copy-pasted. Please use a graphical tool to present homogeneous figures.

We revised Figure 3 to ensure the clarity of the individual panels. We also unified fonts, font sizes and figure colors across all figures.

Figure 1: Structural properties: 0.25 and 0.45 for the different folding of the peptides. What happened to the other 0.3? Is it percent? Please clarify.

In the revised version of the manuscript, we have now clarified that these values represent an estimate of the likelihood of encountering a specific mAA within an α -helix or β -sheet, which does not need to sum to 1. This is because the mAA may adopt other secondary structures, including random coil. We have now clarified this in the caption for Figure 1.

Figure 2c: this should be a table rather than a figure.

We removed it from the figure and introduced it as a new table (Table 3).

Solubility of one compound depends on several things: the medium (solvent, ionic strength, pH...), temperature, and solid form. This has been totally ignored by the authors. Please clarify in the abstract and everywhere else where it appears to be relevant, even in the title I'd expect "Aqueous solubility at x°C".

We are grateful to the reviewer for prompting us to address this important point explicitly in the text. As we now explain more directly in the revised version of the manuscript, our software aims at providing an estimate of the intrinsic solubility of peptides. Since we are focused on relative solubilities, comparing variants of the same peptide, we are not aiming at predicting absolute solubilities and therefore a predictive measure of the intrinsic solubility is sufficient. We have clarified this in the title, abstract and the main body of the text. We acknowledged in particular that our software is correlated with experimental data that was acquired in aqueous solution at room temperature. We also added some further clarification on the term intrinsic solubility and other factors that can affect solubility in paragraph 6.

Abstract:

- *What is the abbreviation "CamSOL-PTM" standing for? Please clarify.*

PTM stands for post-translational modifications. We have added a sentence at the beginning of the abstract to clarify this abbreviation.

- *Please define the chemical space (solvent/T, pH).*

We have clarified this now.

Intro:

- *The authors ignore other available methods to predict solubility. Molecular models that account for the interactions in the liquid phase do exist, which need as input the melting properties of the solids. Such methods are described, e.g., <https://doi.org/10.1039/D0RA08947H>*

We thank the reviewer for pointing this out. At the end of paragraph 5 we mention other solubility predictors. We now added the reference that the reviewer suggested as well. We do not go further into detail on these predictors because to our knowledge there is currently no other sequence-based peptide solubility predictor that can readily handle non-natural amino acids.

Paragraph 5:

- *Also, here, please specify which chemical space.*

- *Please name some methods for state-of-the-art solubility measurements*

Kindly note that in paragraph 5, several advanced protein and peptide solubility measurements are mentioned and referenced (66-69).

Paragraph 6:

You name the peptides PYY, GLP-1, and 18A at the end of paragraph 6. However, in the next paragraph, you start with GLP-1, followed by PYY. Please rephrase and start with 1) PYY, 2) GLP-1, and 3) 18A.

We rephrased the end of paragraph 6.

Results:

- *pKa values calculation: Can you please specify how the pICHemiSt suite calculates the pKa value?*

We now specified how pICHemiSt suite calculates the pKa values of new mAAs.

- *Please enumerate equations*

We have done this now.

- *Equation for p_i^a : In the text, the a_x should be written in italic letters*

We have done this now.

- *experimental solubility: It is not clear which values are presented in Figure 2. Are these values mass% or volume%?*

PEG percentages are mass/volume, i.e. 50 g of PEG was dissolved in buffer giving a final volume of 100 mL. We clarified this in the figure caption as well.

Also, the solubility fraction should have a unit.

The solubility fraction, which represents the proportion of the original material remaining in solution, does not possess a unit of measurement.

Material and Methods:

- *Materials:*

Can the authors comment on properties such as hydrate or solvate formation of the protected amino acids? This is crucial in any solubility experiment, since the solid form critically influences the solubility that is expected. Usually, this is measured by XRD, which is standard in the community of measuring solubility data. For compounds that change their solid form upon precipitation and being contacted by the solvent, the method the authors presented is not valid.

All solubility measurements were obtained on the full peptide sequences, rather than their constituent amino acid building blocks. When produced chemically, peptides are normally obtained as an amorphous powder (i.e. non crystalline material). They will typically be both hydrated, which depends on lyophilisation conditions, and also present as a salt adduct, depending on purification conditions. Therefore, for this work all peptides were synthesised as the same salt form and purified & lyophilised under similar conditions to obtain an amorphous powder with consistent characteristics across different compounds for the experimental solubility measurements.

-*Solubility assay:*

"Solubility was measured with ultracentrifugation". I think, no. For me, it is not clear how solubility can be measured with ultracentrifugation. Is it an optical determination of the solubility? Please specify and add this information to the manuscript. Information on the media that were used, which water solvent, which ph... has to be added. pKa values might be useful to convert all the solubility data to pH 7, but i guess the pH of the samples has not been measured. It will have an effect on solubility.

After centrifuging the samples at pH 7 at 2 mg/mL for 30 min at 100,000 rpm, the peptide concentration remaining in the supernatant was measured using a NanoDrop. At this very fast centrifugation speed any peptide aggregate would pellet down, and what is left in the supernatant is the soluble fraction. This assay only works well for peptides that have low solubility, as otherwise the

supernatant concentration would be just equal to the total concentration (that is nothing aggregates and so nothing pellets down). All measurements were done in the same buffer at pH 7. We rephrased the section to make that clear.

Reviewer: 3

Sequence-based prediction of the solubility of peptides containing non-natural amino acids
In this manuscript, Oeller et al. present a sequence-based method to predict, for the first time, the solubility of peptides containing modified amino acids (mAAs). In short, the authors built a pipeline that predicts the physicochemical properties of any mAAs to incorporate them in the widely used CamSol framework. Solubility predictions for three peptides containing a combination of five different mAAs – for a total of 30 peptide variants – are validated using three different experimental methods.

This is a major update to the CamSol framework. Its novelty, significance and accessibility (through a web server) make it a good fit for the broad audience of Nature Communications, especially those in the fields of biotechnology and drug discovery. The manuscript is clear and does a good job summarizing the current state of the art. I have, however, a few suggestions to strengthen the work and make it suitable for publication:

We thank the reviewer for these positive comments.

Major comments

- Although the method can potentially predict the solubility of peptides containing any mAAs, the experimental validation is performed on mAAs that are of similar size to canonical amino acids. The authors acknowledge this restriction in the discussion, but it should also be mentioned in the abstract and introduction.

We agree with the reviewer about this possible limitation, and have adjusted the abstract and introduction accordingly.

- The peptides chosen by the authors are structurally very similar since all three consist mainly of α -helices, thus ignoring the structural space covered by beta-sheets and beta-strands. This class of secondary structure is structurally very different to α -helices and is more often involved in aggregation events. Furthermore, the authors used the experimental solubilities of the 30 peptide variants to choose, a posteriori, the best combination of parameters for the secondary structure propensity predictor. Since all these peptides consist of α -helices, this could lead to the overfitting of this specific class. To claim the generalizability of the method, we ask to include at least one extra peptide in the validation that consists mainly of beta-sheets, such as a beta-hairpin.

We thank the reviewer for this comment. We appreciate the concern but are confident that our method is not biased towards α -helices, for the following reasons. The reviewer correctly points out that the wild-type forms of the three peptides contain α -helices. However, only the parameters for the secondary structure propensity of mAAs (and – crucially – not of the 20 natural AAs) predictor have been fitted with the new experimental data. None of the other parameters have been changed from previous versions. Therefore, the parameters used to calculate the solubility score for individual amino acids were not changed. Similarly, the parameters defining the calculation for the overall solubility score from the profile were kept unchanged. The original parameters of CamSol were trained on a wide range of proteins containing many different secondary structures and compositions. We added a paragraph to the discussion to clarify this issue.

Furthermore, to make the point for generalizability more convincing we created a new set of 7 PYY variants containing 4 new mAAs (not seen or used previously), measured their solubility and compared it to our predictions. The Pearson's coefficient of correlation for the complete set of PYY variants shows good correlation with 0.6 in line with values obtained on the previous dataset. We are therefore confident that our method is generalizable. The results are depicted in Figures 2 and 3,

table 2 contains the list of new variants and the results and discussion section have been amended accordingly.

Minor comments

- *The CamSolPTM web server needs the SMILES codes of mAAs to predict the solubility of peptides containing non-standard amino acids. It will be useful for potential users to provide a table with the SMILES codes of the most common mAAs.*

We have added a table displaying a list of common mAAs to the webserver.

- *Besides an overall solubility score assigned to the entire sequence, CamSolPTM yields a solubility profile (one score per residue in the protein sequence) where regions with scores larger than 1 denote highly soluble regions, while scores smaller than -1 poorly soluble ones. Previous studies by this group using the CamSol framework have used these per residue values to identify aggregation hot spots. Is the same true for CamSolPTM?*

Yes, this is true for CamSol-PTM. We have highlighted this now also in the discussion.

- *In Figure 3, include in the figure description which variants are removed from the analysis.*

We have now incorporated this information.

REVIEWERS' COMMENTS

Reviewer #2 (Remarks to the Author):

All my comments were considered in the revised version. This version is OK.

It can be accepted, no more comments from my side.

Reviewer #3 (Remarks to the Author):

The authors have addressed all my concerns